# Worldwide genetic diversity of *Plasmodium vivax Pv47* is consistent with natural selection by anopheline mosquitoes

Alvaro Molina-Cruz ●[1] ✉, Lilia Gonzalez-Ceron ●[2], Ankit Dwivedi ●[3], Tran Zen B. Torres ●[1], Nadia Raytselis[1], Micah Young[1], Nitin Kamath[1], Colton McNinch[4], Xinzhuan Su[1], Anthony Ford[5,6,7], Marcelo U. Ferreira ●[8,9], Myriam Arévalo-Herrera[10], Sócrates Herrera ●[10], Eugenia Lo[5], Joana C. Silva ●[3] & Carolina Barillas-Mury ●[1] ✉

Pv47 is the *Plasmodium vivax* ortholog of Pfs47, a protein that allows the *Plasmodium falciparum* malaria parasite to evade mosquito immunity and adapt to diverse vectors. We analyzed global genetic diversity of *Pv47* and compared it with *Pfs47*, finding that most common *Pv47* polymorphisms are non-synonymous and cluster in regions similar to those in *Pfs47*. *Pv47* domain 2 presents an excess of non-synonymous substitutions, suggesting positive selection. The greatest haplotype diversity is found in *Pv47* from East/South-east Asia and Oceania. Like Pfs47, *Pv47* also exhibits a marked geographic population structure worldwide. Notably, a Pv47 polymorphism (K27E) is associated to differences in infectivity to *Anopheles (Nyssorhynchus) albimanus* and *Anopheles pseudopunctipennis*, two phylogenetically distant vectors in Mexico. The striking similarities in genetic diversity, population structure, and signatures of natural selection between *Pv47* and *Pfs47* suggest that adaptation to different *Anopheline* mosquito species drives Pv47 diversity by selecting compatible Pv47 haplotypes.

Malaria remains the most devastating human parasitic disease, with more than 200 M cases and 600,000 deaths annually[1]. Most malaria morbidity (93%) and mortality (95%) are caused by *Plasmodium falciparum* parasites in sub-Saharan Africa, followed by *Plasmodium vivax*, which has a wider geographic distribution[1]. Both *P. vivax* and *P. falciparum* are transmitted by more than 70 anopheline mosquito vectors around the world[2]. Malaria transmission takes place when a mosquito feeds on an infected host. Ingested *Plasmodium* gametocytes transform into gametes in the midgut lumen, and fuse to form a zygote, which gives rise to a motile ookinete. The ookinete must traverse the mosquito midgut epithelium to form an oocyst, which produces thousands of sporozoites that are eventually released into the hemolymph and invade the salivary gland. Sporozoites are transmitted when infected females bite a new vertebrate host[3].

[1]Laboratory of Malaria and Vector Research, National Institute of Allergy and Infectious Diseases, National Institutes of Health, Rockville, MD, USA. [2]Centro Regional de Investigación en Salud Pública, Instituto Nacional de Salud Pública, Tapachula, Chiapas, México. [3]Institute for Genome Sciences, University of Maryland School of Medicine, Baltimore, MD, USA. [4]Bioinformatics and Computational Biosciences Branch, Office of Cyber Infrastructure and Computational Biology, National Institute of Allergy and Infectious Diseases, National Institutes of Health, Bethesda, MD, USA. [5]Department of Biological Sciences, University of North Carolina at Charlotte, Charlotte, NC, USA. [6]Department of Bioinformatics and Genomics, University of North Carolina at Charlotte, Charlotte, NC, USA. [7]Center for Computational Intelligence to Predict Health and Environmental Risks, University of North Carolina at Charlotte, Charlotte, NC, USA. [8]Department of Parasitology, Institute of Biomedical Sciences, University of São Paulo, São Paulo, Brazil. [9]Global Health and Tropical Medicine, Associate Laboratory in Translation and Innovation Towards Global Health, Institute of Hygiene and Tropical Medicine, NOVA University of Lisbon, Lisbon, Portugal. [10]Malaria Vaccine and Drug Development Center, Cali, Colombia. ✉e-mail: amolina-cruz@niaid.nih.gov; cbarillas@niaid.nih.gov

Pv47 is the *P. vivax* ortholog of Pfs47, a protein on the surface of ookinetes that greatly enhances malaria transmission by allowing *P. falciparum* to evade the mosquito immune system. The interaction of Pfs47 with a compatible mosquito midgut receptor prevents activation of a caspase-mediated nitration response in ookinete-invaded epithelial cells[4–7]. Disruption of this nitration response, in turn, prevents elimination of the ookinete by the mosquito complement-like system. Pfs47 is a member of the 6-cysteine (6-Cys) protein family, with three characteristic "P48-45 domains", that has orthologs in all *Plasmodium* species. The first and third domains are canonical P48-45 domains with six Cys that form three disulfide bridges, whereas the second domain (Pfs47-D2) is more flexible, with only two Cys, predicted to form a single disulfide bridge[8]. Pfs47 is genetically diverse across populations worldwide and polymorphisms between the two Cys in the second domain (Pfs47-D2) determine compatibility to evolutionary distant mosquito vectors[9–11]. The dramatic geographic population structure of Pfs47 supports the working hypothesis that Pfs47 is important for *P. falciparum* to adapt to different mosquito vectors[9]. *P. falciparum* malaria originated in Central Africa and dispersed as humans migrated around the world[12]. Those *P. falciparum* parasites with a Pfs47 haplotype compatible with the different anopheline species present in new geographic regions were effectively transmitted, resulting in natural selection of the parasites circulating in different geographic areas[12]. Furthermore, antibodies to a specific region of Pfs47-D2 can disrupt *P. falciparum* transmission, making Pfs47 a potential target antigen for a transmission blocking vaccine[13].

The widespread geographical distribution of *P. vivax* suggests this parasite also had to adapt to different mosquito species around the world to be transmitted locally. In fact, genetic analysis of *P. vivax* populations in Southern Mexico, as well as experimental mosquito infections, revealed a parasite population structure that appears to be the result of natural selection by two evolutionary distant mosquito species, *An. albimanus* (subgenus *Nyssorhynchus*) in the lowlands and *An. pseudopunctipennis* (subgenus *Anopheles*) in the foothills[14]. In addition, genome-wide analysis of *P. vivax* revealed that *Pv47* is one of the genes with the strongest population structure between *P. vivax* from South America and Southeast Asia[15], suggesting that Pv47 is under strong natural selection. The Pv47 ortholog in *Plasmodium simium*, which infects Neotropical platyrrhine monkeys and derived recently from human *P. vivax*, also presents polymorphisms that correlate to transmission by different sylvatic anopheline vectors of the subgenus *Kerteszia*[16]. Furthermore, Pv47 is a viable transmission blocking vaccine target[17], suggesting that it plays a role during mosquito infection. Here, we analyze the genetic diversity and evolution of *Pv47* worldwide and compare it to *Pfs47* to further study the potential role of Pv47 in vector compatibility and adaptation of *P. vivax* parasites to different mosquito species.

## Results

### Worldwide genetic diversity and population structure of *Pv47*
The genetic diversity of *Pv47* (PVX_083240; PVP01_1208000.1) was established by analyzing 1199 Pv47 gene coding sequences (Supplementary Data 1) obtained in 28 countries worldwide (Table S1). In total, 71 polymorphic sites were identified in the data set, most of which (68%) contained non-synonymous mutations (Table S2). A total of 209 *Pv47* haplotypes were identified (Supplementary Data 2), the majority only present in a specific geographic area (Fig. 1A). Most of the *Pv47* gene sequence is highly conserved, with amino acid sequence similarity between haplotypes ranging from 95% to 99.8%. However, there are clusters of non-synonymous polymorphisms close to the predicted N-terminus of domain 1 (Pv47-D1) and in the second domain (Pv47-D2), and one polymorphism is also frequently observed in the third domain (Pv47-D3) (Fig. 1B, left panel). Sites with nucleotide diversity ($\pi$) >0.05 were found to be non-synonymous substitutions (Fig. 1B).

*Pv47* presented high haplotype diversity (Hd) in the overall population (Hd = 0.95; Table S3). Interestingly, *Pv47* sequences from East Asia/ Southeast Asia and Oceania had the highest haplotype diversity, Hd = 0.94 and Hd = 0.93, respectively, and the largest average genetic distance between sequence pairs (estimated as nucleotide diversity, $\pi$), with $\pi$ = 0.0026 and $\pi$ = 0.0035, respectively (Table S3). The rate of nonsynonymous polymorphisms per site was significantly higher than the rate of synonymous polymorphisms in the Pv47-D2, suggestive of natural selection, especially in South Asia, East Asia/ Southeast Asia and Oceania's populations (Table S3). At least three regions of *Pv47*, associated with non-synonymous polymorphisms in Pv47-D1, D2 and D3, presented Tajima's D > 2, also suggestive of natural selection (Fig. S1A).

The frequency of the most common Pv47 protein haplotypes (frequency > 0.05) (Figs. 1C, S2) differed widely between geographic regions. Out of 71 polymorphic sites in the Pv47 coding region, 17 SNPs presented population structure with $F_{ST} > 0.05$, and 12 non-synonymous SNPs (causing residue changes F22L, F24L, K27E, S57T, S62N, L82V, D156G, V230I, M233I, I262K/T, I273M/V, A373V) had $F_{ST} > 0.2$ between some of the continental populations analyzed (Fig. 1C).

While most Pv47 sequences from South America (75.7%) were identical to the reference Pv-Sal I (from a strain collected in El Salvador), alternate Pv47 haplotypes F22L and K27E (Pv47-D1) were frequent (>0.7) in Mexico and other regions of the world. Alternate haplotypes F24L (Pv47-D1) and I262K/T (Pv47-D2) were also frequent beyond the Americas, while M233I (Pv47-D2) was common in South and Southeast Asia, and A373V (Pvs47-D3) was common in Africa, Southeast Asia, and Oceania. Allele encoding V230I and I273V (Pv47-D2) was frequent in Southeast Asia, while haplotype I273M (Pv47-D2) was frequent in Africa and Oceania (Figs. 1C, S2).

In general, haplotype network analysis shows separation of *Pv47* haplotypes circulating in different continents (Fig. 2A), except for Africa, where the most frequent haplotypes are shared with Oceania or present at low frequency in Southeast Asia (Fig. 1C). A marked population structure was also obtained for *P. falciparum* Pfs47 (Table S5), consistent with previous reports[9,18–20], which correlates with the different anopheline mosquito species that transmit malaria in a given region[9].

The largest genetic distances in *Pv47* sequences were between populations from South America and those from Africa ($F_{ST}$ 0.76–0.83), South Asia ($F_{ST}$ 0.67–0.86), Southeast Asia ($F_{ST}$ 0.63–0.86) and Oceania ($F_{ST}$ 0.60–0.70) (Fig. 2B). There were also significant genetic differences between Middle East/South Asia and Southeast Asia ($F_{ST}$ 0.25–0.66), and between Oceania and Middle East/South Asia ($F_{ST}$ 0.39–0.50) and Southeast Asia ($F_{ST}$ 0.28–0.42). Interestingly, significant genetic differences were also found within the New World, between South America and Mexico ($F_{ST}$ 0.38–0.64) (Fig. 2B).

### Worldwide genetic diversity and population structure of *Pfs47*
Detailed genetic diversity analysis of 4971 *Pfs47* (PF3D7_1346800) sequences confirmed previous reports[20,21] indicating that the most frequent non-synonymous polymorphisms in *Pfs47* localize close to the predicted N-terminus of domain 1 (Pfs47-D1) and the second domain (Pfs47-D2), while a polymorphism in the third domain (Pfs47-D3) is less frequent (Fig. 1B, right panel). *Pfs47* also presented high haplotype diversity in the overall population (Hd = 0.89; Table S4), with the highest haplotype diversity within South Asia (Hd = 0.87, Table S4). Notably, the average genetic distance between sequences from PNG was particularly large ($\pi$ = 0.0022, Table S4).

The largest genetic differentiation in *Pfs47* was between populations from South America and Asia ($F_{ST}$ 0.79–0.94) followed by South America and Africa ($F_{ST}$ 0.72–0.94) and Africa and Asia ($F_{ST}$ 0.73–0.88) (Table S5). Significant population differences were also found within continents, for example, population differentiation of *Pfs47* increased

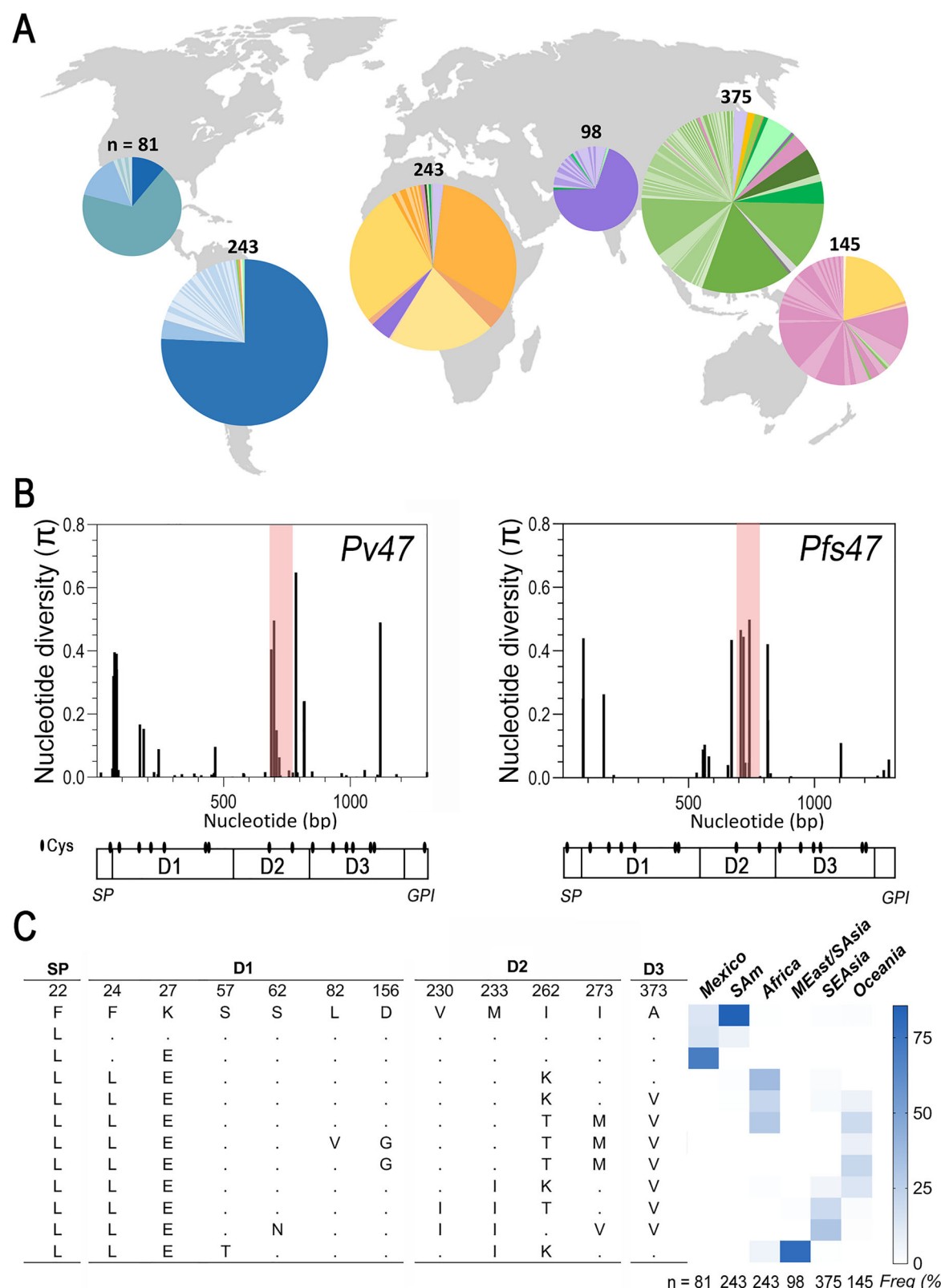

from West to East Africa and Madagascar ($F_{ST}$ 0.32-0.38) (Table S5); while Central Africa (DRC) presented a modest difference with the rest of Africa ($F_{ST}$ 0.09-0.25). In Asia, a significant population structure was observed between Southeast Asia and South Asia (Bangladesh)/Oceania (PNG) ($F_{ST}$ 0.45–0.53), while in South America, *Pfs47* presented marked population difference across the Andes, between the coastal regions of Colombia/Central America and the Amazonia in French

Guyana/Peru/Brazil ($F_{ST}$ 0.51–1.0; Table S5). Four regions of *Pfs47*, associated with non-synonymous polymorphisms in Pfs47-D1 and D2 presented Tajima's D > 2, suggesting balancing selection, although other causes are possible (Fig. S1B).

Several *Pfs47* amino acid polymorphisms differed between geographic regions. *Pfs47* haplotypes private to South America differed from African haplotypes in non-synonymous polymorphisms in Pfs47-

**Fig. 1 | Geographic distribution of Pv47 haplotypes and genetic diversity of Pv47 and Pfs47. A** Distribution of Pv47 protein haplotypes across different geographic regions. Haplotype frequency is represented by the size of the slice area in a pie chart. The color of each haplotype corresponds to the region where it is most frequent: South America is represented in blue; Mexico, Aqua; Africa, orange; Middle East/South Asia, purple; Southeast Asia, green; Oceania, pink. The number of *Pv47* sequences analyzed per region are indicated. **B** Nucleotide diversity (π) per bp for *Pv47* and *Pfs47* gene coding sequences. The predicted protein domain structure and cysteine (Cys) locations (ovals) are indicated for both *Pv47* and *Pfs47*.

SP, signal peptide; D1, domain 1; D2, domain 2; D3, Domain 3; GPI, predicted glycosylphosphatidylinositol anchoring region. The region between two Cys in D2 is shaded in red. Source data are provided as a Source Data file 1. **C** Frequency of Pv47 aa haplotypes of polymorphisms with strong geographic structure. The Pv47 haplotypes shown are defined by SNPs that exhibit a marked population structure ($F_{ST} > 0.2$) between any of the regions compared and have a frequency greater than 5% in at least one of the geographic regions analyzed. SAm South America, MEast Middle East, SAsia South Asia, SEAsia Southeast Asia.

D2 (T236I, S242L, V247A, and I248L) which are nearly fixed between continents. These are polymorphisms that have been previously shown to be important for mosquito immune system evasion and parasite compatibility with anophelines[9–11]. Asian private *Pfs47* haplotypes also differed from African haplotypes mostly in polymorphisms altering residues in Pfs47-D2 (I224N, T236I, L240I, I248L and N272Y) and one in Pfs47-D1 (L28I) (Fig. S3).

Within Africa, the frequency of the alternate *Pfs47* haplotype E27D was high in East Africa, while in Central Africa haplotypes with E188D and N272Y/I were more frequent. In the New World, there was a major difference in *Pfs47* populations between Colombia and Amazonia, with the non-reference haplotype I178V being fixed in coastal areas of Colombia, while the reference genotype encoding T68M was fixed in Amazonia. In Asia, there were also significant genetic differences in *Pfs47* between Southeast Asia and Bangladesh, mostly due to differences in polymorphisms L28I, E55K, L240I and N272Y bp (Fig. S3).

### Experimental evidence of selection of Pv47 by anopheline vectors

*An. albimanus* is the main vector of *P. vivax* in the lowlands of Chiapas, Mexico, while *An. pseudopunctipennis* is the main vector in the piedmont of the *Sierra Madre* mountain range[14]. Previous studies showed that the geographic distribution of three genetically distinct *P. vivax* populations correlates with the geographic distribution of these two vectors. Furthermore, side-by-side experimental infections in which both mosquito vectors were fed on blood from the same infected human host showed that genetically distinct *P. vivax* populations differed in their ability to infect these two mosquito vectors[14,22]. The authors suggested that the observed *P. vivax* population structure in Chiapas could be explained by differences in compatibility with these two vector species.

We investigated whether polymorphisms in *Pv47* could explain the observed differences in vector compatibility by genotyping *Pv47* in 43 isolates previously tested in experimental infections[14] in which the infection prevalence was at least 2-fold higher in one of these two mosquito species and the difference in infection prevalence was statistically significant (Chi-square, $p < 0.05$) (Table S6). Of these *P. vivax* isolates, 15 had higher infections in *An. albimanus*, while 28 infected *An. pseudopunctipennis* more efficiently (Fig. 3A, Table 1, Table S6). A total of six *Pv47* haplotypes were identified (Table 1). Three non-synonymous polymorphisms were present in single isolates (2%), while two of them F22L (88%) and K27E (65%), were frequent (Table 1). The K27E polymorphism in Pv47-D1, in proximity to the predicted N-terminal of Pv47, had a perfect correlation with the differences in vector compatibility. Those isolates with a positively charged lysine (K27) in this position had a significantly higher infection intensity in *An. albimanus* (Mann-Whitney; ****, $P < 0.001$), while those with a negatively charged glutamic acid (E27) had a significantly higher intensity of infection in *An. pseudopunctipennis* (Mann-Whitney; ****, $P < 0.001$), (Table 1, Fig. 3A).

### Predicted molecular structure of Pv47 and Pfs47

In silico modeling of *Pv47* and *Pfs47* proteins using Alphafold2[23] predicted very similar structures despite having a modest level of amino acid identity (43%) (Fig. 3B). Both proteins consist of three 6-Cys s45/

48 domains[8] with a characteristic ß-sandwich fold formed by anti-parallel and parallel ß-sheets (Fig. 3B). Domains D1 and D3 have the canonical 6-Cys pattern. In contrast, D2 is a shorter and degenerate s48/45 domain with only two cysteines, which is linked to the other two domains by flexible, less organized regions. Interestingly, the major polymorphisms in Pv47-D1, close to its N-terminal end, are predicted to be in spatial proximity to the protein surface where the major polymorphisms in Pv47-D2 are present. The major polymorphisms in Pfs47-D2 are known to be critical for immune evasion of the mosquito immune system and are major determinants of compatibility with different mosquito vector species through interaction with the Pfs47 receptor in the mosquito gut[6,9,10]. The predicted 3-D structure suggests that D1 amino acid polymorphisms in Pv47 -including K27E- may also interact with the mosquito Pfs47Rec to mediate immune evasion.

## Discussion

The worldwide genetic diversity of *Pv47* presents important similarities with that of *Pfs47*, suggesting that *Pv47* is also under selection by different mosquito vectors. In both *Pv47* and *Pfs47*, common polymorphisms (>5%) are mostly non-synonymous, and they are non-randomly distributed (Fig. 1B). In addition, most substitutions in Domain 2 of *Pv47* and *Pfs47* have signatures of positive selection (Tables S3, S4). This is notable as four polymorphisms between the two Cys of Domain 2 of Pfs47 were previously shown to determine compatibility to different anopheline species[9]. Both *Pv47* and *Pfs47* have a strong geographic population structure, with the greatest genetic distances between South American populations and those from other continents (Fig. 2B, Table S5). Particularly interesting is the high nucleotide diversity of both genes in Oceania, a region far away from the presumed African center of origin of *P. falciparum* and *P. vivax*[24]. These common aspects suggest that these orthologs share genetic features resulting from a common function and of similar natural selection pressure. There is also significant geographic structure in both *Pv47* and *Pfs47* within continents that corresponds with regional differences in mosquito vector species, consistent with a vector-driven selection of both genes. For example, Pv47 and Pfs47 haplotypes from South Asia (Bangladesh), where the dominant vectors are *Anopheles stephensi* and *Anopheles culicifacies*, differ genetically from Southeast Asia, where the dominant vectors are *Anopheles dirus* and *Anopheles minimus*[2]. The strong population structure seen in *Pv47* worldwide makes it a good potential marker to develop assays to identify the geographic origin and transmission potential of imported malaria cases, as has been developed for *Pfs47*[20]. Pv47 haplotypes from the New World differ significantly from the other continents by polymorphism F24L, while African Pv47 haplotypes differ significantly from Asia and Oceania by the polymorphism M33I.

South American *Pv47* and *Pfs47* sequences exhibit the lowest genetic diversity among populations across the globe. The lower diversity may be in part due to a population bottleneck during the recent arrival (within the last 500 years) of *P. vivax* and *P. falciparum* on the American continent. However, it is estimated that more than 6 M Africans from different regions were brought into the new world as part of the slave trade, so it is likely that they were infected with genetically diverse *P. falciparum* parasites[12]. In addition, strong selection of parasites with African *Pfs47* haplotypes by *An. albimanus*, a

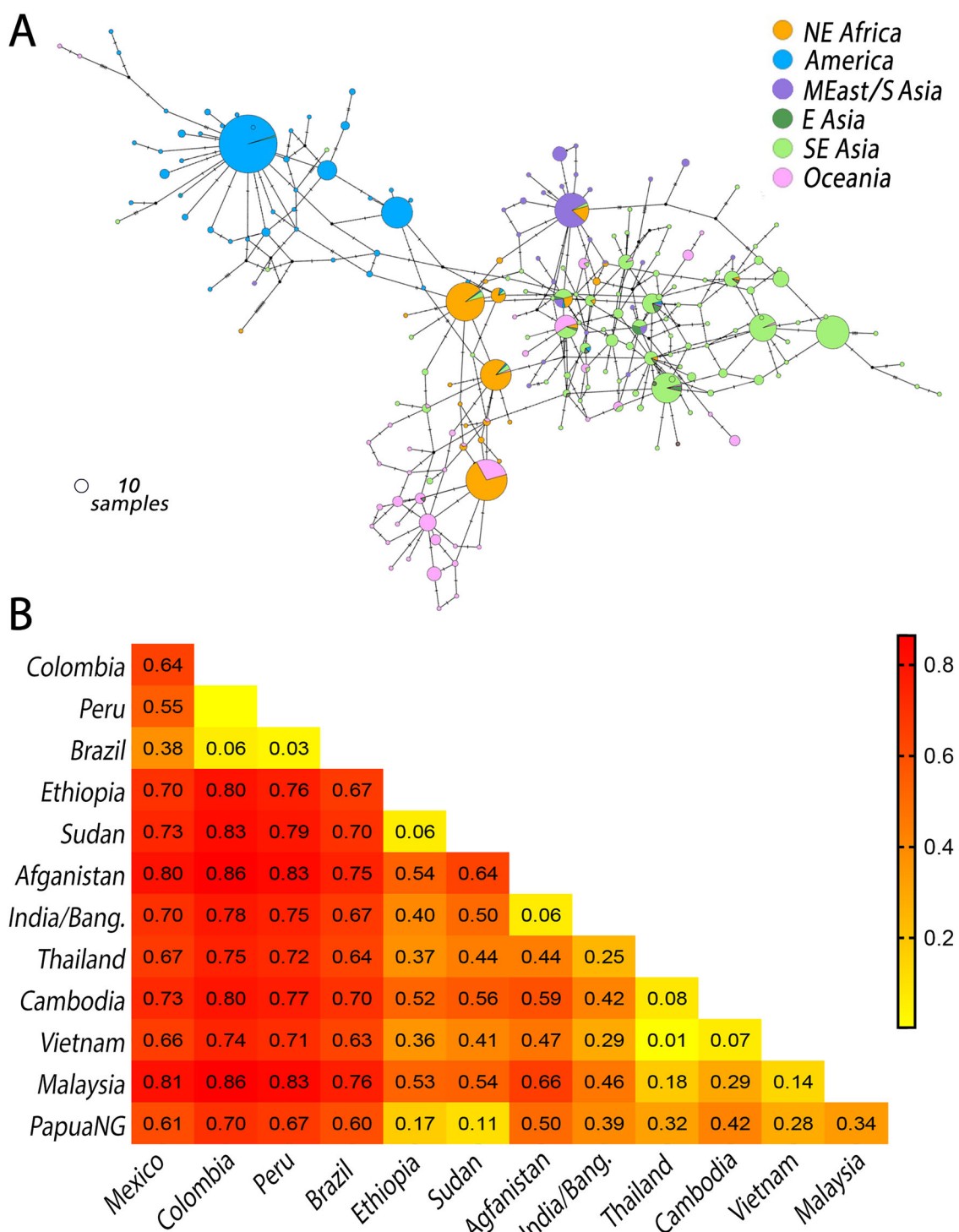

**Fig. 2 | Genealogy and population structure of *Pv47* haplotypes. A** Haplotype network (TCS) of the 209 *Pv47* haplotypes identified world-wide. The geographic origin of each haplotype is indicated by a different color. The size of the circular node representing each haplotype is proportional to the number of samples with that sequence (a circle representing 10 samples is shown as reference). The perpendicular marks on the branches between haplotypes indicate the number of nucleotide substitutions separating the two haplotypes. NE Africa, Northeast Africa; America, Mexico and South America; MEast/S Asia, Middle East and South Asia; E Asia, East Asia; SE Asia, Southeast Asia. Source data are provided as a Source Data file 2. **B** Fixation index ($F_{ST}$) among *Pv47* populations analyzed.

major South American vector evolutionarily distant from African anophelines, was experimentally demonstrated[9]. This indicates that the low genetic diversity of *Pfs47* in South America is likely due to strong vector-driven selection of compatible Pfs47 haplotypes. However, *P. vivax* malaria is rare in Africa because the Duffy mutation, which protects from *P. vivax* infection, is widespread in African

populations. Recent evidence indicates that *P. vivax* malaria was initially brought to the Americas mainly by Europeans[25]. This could explain, in part, the lower genetic diversity of *Pv47* in the Americas. Nevertheless, evolutionarily distant vectors in the American continent may have also exerted strong selection of compatible *Pv47* haplotypes, as observed in Mexico.

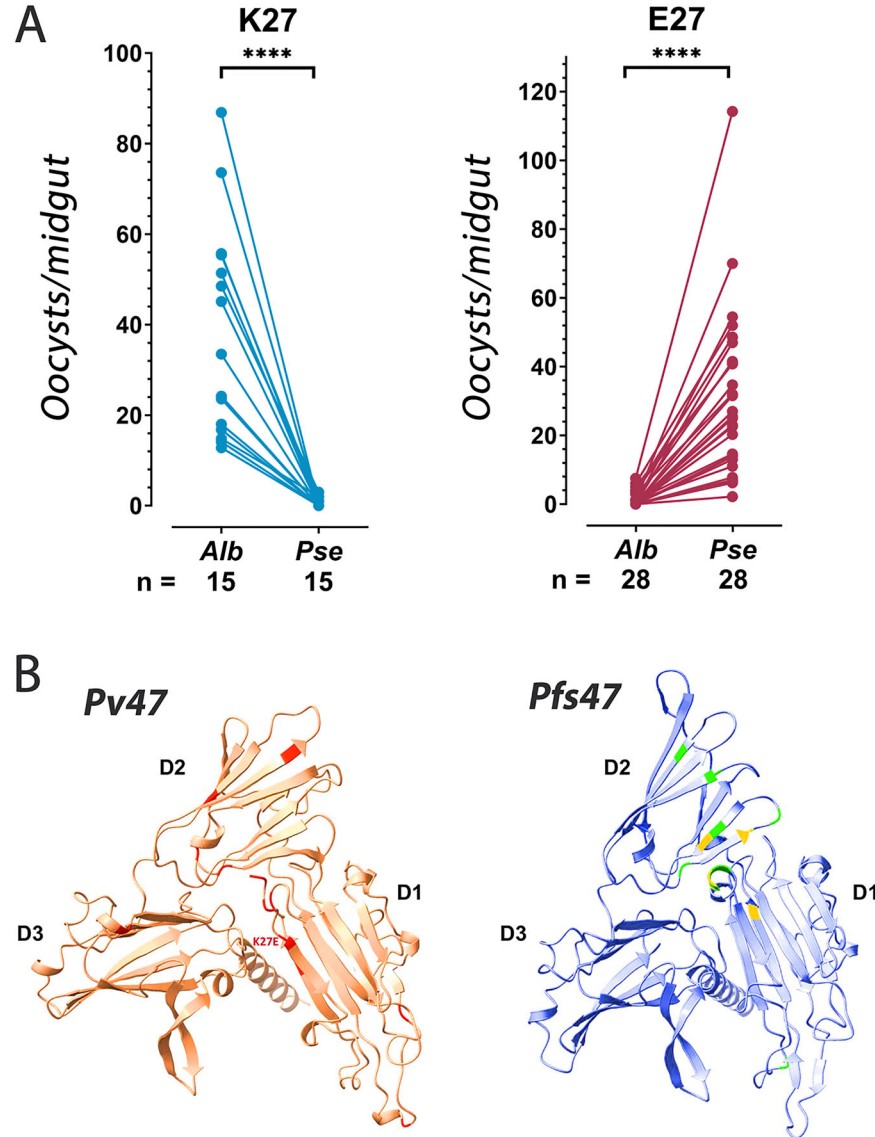

**Fig. 3 | Association of the Pv47 haplotype polymorphism K27E with differential infectivity of two anopheline vectors. A** Average number of *P. vivax* oocysts per midgut in *Anopheles albimanus* (Alb) and *Anopheles pseudopunctipennis* (Pse) mosquitoes infected with blood from the same individual infected with *P. vivax* parasites in the field that carry either the K27 or the E27 amino acid polymorphisms. The number of independent infections is indicated; two-tailed t-test; ****, p < 0.0001. Source data are provided as a Source Data file 1. **B** Protein structure of Pv47 predicted by Alphafold2 (left). The position of amino acid polymorphisms that present strong population structure ($F_{ST}$ > 0.5; >0.05 frequency) are indicated in red color. Predicted location of amino acid polymorphism K27E is indicated. Protein structure of Pfs47 predicted by Alphafold2 (right). The position of amino acid polymorphisms that were previously shown to determine vector compatibility are indicated in light green and yellow. Predicted protein domains are indicated (D1, D2, D3).

It is remarkable that both *Pv47* and *Pfs47* have high genetic diversity in Oceania (Tables S3 and S4), a region that is not considered to be the center of origin of either of the two parasites. This suggests that either the malaria vectors in Oceania have low *P47* specificity and do not select for specific haplotypes of either protein enabling multiple mutations to emerge and persist in this region, or alternatively, that the high diversity of malaria vectors in Oceania, each with its own specificity, results in the local persistence and continual transmission of parasites with different *Pv47* and *Pfs47* haplotypes. The exceptionally high levels of malaria transmission in Oceania, the highest after Africa[1], could also be related to the observed high genetic diversity of *Pv47* and *Pfs47*.

There are also some important differences between the population genetics of *Pv47* and *Pfs47*. In general, *Pv47* has a slightly higher genetic diversity, measured as haplotype and nucleotide diversity, than *Pfs47* (Tables S3 and S4). This is consistent with the overall greater genetic diversity in the *P. vivax* genome compared to that of *P. falciparum*, presumably because *P. vivax* is a parasite that has been infecting humans for a longer time than *P. falciparum*. Selection of *Pv47* by vectors could also lead to higher diversity of *Pv47*, since *P. vivax* has a broader ecoregional range due to its ability to survive at lower temperatures than *P. falciparum*[26]. Furthermore, *Pfs47* has a larger genetic distance between African and Asian populations than *Pv47*. The origin of *P. vivax* malaria currently present in Africa is a topic of debate, and the parasite may have been re-introduced from other continents, after it was almost eradicated from Africa by the protective effect of the Duffy mutation, while *P. falciparum* malaria persisted in Africa. However, some of the most frequent *Pv47* haplotypes in Africa are only found at a low frequency in Oceania and Asia (Figs. 1A, 2A), suggesting that they may represent haplotypes that have been selected by African anopheline vectors.

**Table 1 | Pv47 haplotypes in Southern Mexico and their infectivity to *Anopheles albimanus* and *Anopheles pseudopunctipennis***

| Sample | Pv47 Polymorphisms (bp/aa) | | | | | Preferential vector | n |
|---|---|---|---|---|---|---|---|
| | 66 F22L | 79 K27E | 255 V85V | 1189 E397K | 1282 A428T | | |
| Sal-I (Reference) | C | A | G | G | G | | |
| MEX-009 | A | . | . | . | . | *An. alb.* | 12 |
| MEX-036 | . | . | . | . | . | " | 25 |
| MEX-051 | . | . | . | . | . | " | 25 |
| MEX-053 | A | . | . | . | . | " | 25 |
| MEX-054 | A | . | . | . | . | " | 25 |
| MEX-055 | A | . | . | . | A/G | " | 25 |
| MEX-059 | A | . | . | . | . | " | 25 |
| MEX-060 | . | . | . | . | . | " | 25 |
| MEX-061 | . | . | . | . | . | " | 25 |
| MEX-063 | A | . | . | . | . | " | 25 |
| MEX-065 | A | . | . | – | – | " | 25 |
| MEX-136 | A | . | . | . | . | " | 10 |
| MEX-158 | A | . | . | . | . | " | 10 |
| MEX-223 | A | . | . | . | . | " | 25 |
| MEX-255 | . | . | . | . | . | " | 25 |
| MEX-049 | A | G | . | . | . | *An. pseudo.* | 25 |
| MEX-056 | A | G | . | . | . | " | 25 |
| MEX-062 | A | G | . | . | . | " | 25 |
| MEX-098 | A | G | . | . | . | " | 25 |
| MEX-112 | A | G | . | . | . | " | 25 |
| MEX-120 | A | G | . | . | . | " | 25 |
| MEX-122 | A | G | . | . | . | " | 25 |
| MEX-141 | A | G | . | . | . | " | 10 |
| MEX-144 | A | G | . | . | . | " | 25 |
| MEX-145 | A | G | . | . | . | " | 9 |
| MEX-178 | A | G | . | – | – | " | 25 |
| MEX-179 | A | G | . | . | . | " | 25 |
| MEX-199 | A | G | . | . | . | " | 25 |
| MEX-201 | A | G | . | . | . | " | 25 |
| MEX-202 | A | G | . | . | . | " | 25 |
| MEX-208 | A | G | . | . | . | " | 25 |
| MEX-209 | A | G | . | . | . | " | 25 |
| MEX-211 | A | G | A | . | . | " | 25 |
| MEX-214 | A | G | . | . | . | " | 25 |
| MEX-216 | A | G | . | A/G | . | " | 25 |
| MEX-218 | A | G | . | . | . | " | 25 |
| MEX-235 | A | G | . | . | . | " | 15 |
| MEX-241 | A | G | . | . | . | " | 9 |
| MEX-248 | A | G | . | . | . | " | 10 |
| MEX-249 | A | G | . | . | . | " | 25 |
| MEX-251 | A | G | . | . | . | " | 25 |
| MEX-253 | A | G | . | . | . | " | 25 |
| MEX-254 | A | G | . | . | . | " | 25 |

The Pv47 polymorphism K27E, which correlates with differences in vector compatibility is indicated (gray shading).

Another interesting difference between *Pv47* and *Pfs47* is that *Pfs47* haplotypes in the coastal region of Colombia, where the dominant vector is *An. albimanus* differ from those in Amazonia, where the dominant vector is *Anopheles (Nyssorynchus) darlingi* (Table S5). This would suggest that the haplotype compatibility of *Pfs47* is sufficiently different between these two *Nyssorynchus* vectors to represent a barrier to gene flow between the two regions. The existence of such a barrier may explain why the chloroquine resistance that arose in South America has not spread to Central America[27]. However, *Pv47* does not appear to have a similar population structure between the coastal regions and Amazonia of South America.

The Pv47 from Mexican isolates with different infections in *An. albimanus* compared to *An. pseudopunctipennis*, differs in a polymorphism (K27E) (Table 1); this polymorphism is predicted to be in proximity to the protein surface region in Pfs47-D2 which, in turn, determines vector compatibility (Fig. 3B). This is consistent with the selection of *Pv47* by these two evolutionary distant vectors and does not align exactly with the *P. vivax* population structure described by Joy, et al. in that region[14]. While all the subpopulation C1 parasites from the coastal area carry the K27 allele and resulted in higher infection in *An. albimanus*, both subpopulation F1 and F2 parasites share the alternate allele, E27 (Table S6). The K27E polymorphism in Pv47 is fixed to the non-reference allele (E) in all continents aside from South America. Interestingly, the E27D polymorphism in Pfs47 varies in frequency from West Africa to East Africa[28], suggesting that E27D is also associated with differences in vector compatibility.

Other *P. vivax* ookinete-specific genes important for mosquito midgut infection, such as Pvs25, Pvs28, SOAP and chitinase, also present polymorphisms that correlate with differential vector compatibility in Southern Mexico[29–31]. Although it is possible that these genes are selected by different vectors in that region, none of them exhibit a striking population structure worldwide, as has been shown for Pv47[15].

The recent transfer of human *P. vivax* to platyrrhine monkeys in Southeastern Brazil that gave rise to *P. simium* also involved adaptation to transmission by anopheline vectors of the *Kerteszia* subgenus, mainly *Anopheles cruzii* and *Anopheles bellator*. Coincidentally, this case of reverse zoonosis strongly correlates with two polymorphisms in *P. simium* P47 (V129L, D194G) (Fig. S4)[16] that were only detected at low frequency in Pv47 Brazilian populations (Fig. S5). Polymorphism V129L is predicted to be on the same surface of Pv47 and in close proximity to K27E, suggesting that it is involved in determining vector compatibility.

The *P. vivax*-like parasites that infect primates in central Africa are the closest known relatives to *P. vivax*. An analysis of 10 *P. vivax*-like *P47* sequences recovered from chimpanzees and one mosquito[32] revealed higher sequence similarities to Pv47 haplotypes circulating in Africa and Oceania (haplotype "e" in Fig. S4; and red circles in Fig. S5), including polymorphisms with strong population structure in *Pv47* ($F_{ST} > 0.2$; Fig. S4). This suggests that the ancestral *P. vivax*, presumably transmitted by sylvatic ape malaria vectors (e.g., *An. moucheti*), could have readily adapted to human malaria vectors in Africa and Oceania.

Overall, our population genetic analysis and vector compatibility studies support the hypothesis that *Pv47* allows the parasite to evade the mosquito immune system, mediating adaptation of *P. vivax* to evolutionary distant vectors worldwide by determining their compatibility with a given vector and, thus, their transmission potential.

## Methods
### *Pv47* gene sequences
A total of 1199 *Pv47* world-wide sequences were analyzed. Sequence sources included GenBank (71 sequences), PlasmoDB (99 sequences), the Malaria Genomic Epidemiology Network (MalariaGEN)

(794 sequences) the University of North Carolina (112 sequences) (GenBank accession nos. PV959330 - PV959441) and the European Nucleotide Archive (24 sequences) (Project PRJEB4409[33] accession nos. ERR775189-92, ERR925409-12, ERR925416-7, ERR925420-1, ERR925424, ERR925430-1, ERR925433-41). *Pv47* sequences from MalariaGEN include only sequences with at least 3 readings coverage, and at most one polymorphism to be able to reconstruct haplotypes unequivocally. Some *Pv47* sequences from Mexico, Colombia and Brazil were obtained from PCR products from DNA extracted from field samples (GenBank accession nos. PV606815–PV606913) (99 sequences). *Pv47* PCR was done with nested PCR in three regions of this intronless gene. The 5′ region was first PCR amplified in a 20 μl reaction (95 °C for 5 min then 25 cycles of 55 °C for 2 min, 72 °C for 2 min, and 94 °C for 1 min; ending with 55 °C for 2 min and 72 °C for 5 min) using primers Pv47A_ExF (TGTAAGTGCCTGCCTACCAA), Pv47A_ExR (TCCTTCACCTTCACCAGTTTG). One microliter of the first reaction was used in a 20 μl PCR (95 °C for 5 min then 30 cycles of 55 °C for 2 min, 72 °C for 2 min, and 94 °C for 1 min; ending with 55 °C for 2 min and 72 °C for 5 min) using nested primers Pv47A_InF (GACCACTAAACGGGAAGTGC), Pv47A_InR (TGTGAAATCGATTCCTCTTGG). The middle region of Pv47 was amplified in similar manner using primers Pv47B_ExF (TACTGCCGATGCGACAATAG), Pv47B_ExR (TCTTGCTGGAGCCGATATGT), and then nested primers Pv47B_InF (AAAGGGGAGGACCAAGAAAA), Pv47B_InR (CTAAGGCAAACCCATCTTGG). The 3′ region of Pv47 was amplified in similar manner using primers Pv47C_ExF (AAGTGCAAAAACACCTGTAAGGA), Pv47C_ExR (AGCATGCTGCCCTAATCATC), and then nested primers Pv47C_InF (AAGACGCAGAAAACGGAAAA), Pv47C_InR (CCACACATGTGGCTATCTGC). This publication uses data from the MalariaGEN *Plasmodium vivax* Genome Variation Project as described[34].

### *Pfs47* gene sequences

A total of 4971 *Pfs47* world-wide sequences were used. Sequences were obtained from the literature[9,35] and the Malaria Genomic Epidemiology Network (Malaria-GEN) (25) as described previously[20]. In the case of the *Pfs47* sequences in samples from other sources, only the major allele was analyzed as described[35].

This publication uses data from the MalariaGEN *P. falciparum* Community Project, PfCP (www.malariagen.net/projects/p-falciparum-communityproject) and the Pf3K project (2016) pilot data release 5. Genome sequencing was performed by the Wellcome Trust Sanger Institute and the Community Projects, coordinated by the MalariaGEN Resource Centre with funding from the Wellcome Trust (098051, 090770).

### Genetic diversity and network analysis of *Pv47* and *Pfs47* gene sequences

The genetic diversity analyses of Pv47 and Pfs47 were carried out by gene and by domain with DnaSP6[36] and MEGAX[37]. DNA gene sequences were aligned with the CLUSTALW algorithm in MEGAX. The number of polymorphic sites (S), number of haplotypes (H), haplotype diversity (Hd), and Tajima's D statistic were estimated with DnaSP6. Nucleotide diversity (average number of nucleotide substitutions in pairwise sequence comparisons, π) was estimated with MEGAX's distance estimation using the Jukes-Cantor model for nucleotide substitutions, uniform rates among sites and 1000 bootstrap replications.

Evidence of natural selection was inferred with MEGA6 by estimating the difference between the average number of synonymous substitutions per synonymous site (dS) and non-synonymous substitutions per non-synonymous site (dN) in pairwise sequence comparisons, using the Nei-Gojobori method and Jukes-Cantor correction. One thousand bootstrap replications were used to estimate its standard error and two tailed Z-test for the null hypothesis that polymorphisms were not under selection (dS = dN).

DNA network analysis (TCS) was done with PopArt[38] with a nexus file generated with DnaSP6.

The fixation index ($F_{ST}$) was estimated in pairwise comparisons among *Pv47* populations that had at least 23 sequences using nucleotide-based statistics[39] in DnaSP6.

### In silico modeling of *Pv47* and *Pfs47* proteins

The three dimensional structure of proteins for the Pv47 reference haplotype Sal I and the Pfs47 reference haplotype 3D7 were predicted with Alphafold2[23] software using the Alphafold server (https://alphafoldserver.com/). The PDB files of the highest ranked protein models were visualized and edited using ChimeraX 1.2.5 software[40] (https://www.rbvi.ucsf.edu/chimera/).

### Reporting summary

Further information on research design is available in the Nature Portfolio Reporting Summary linked to this article.

## Data availability

Pv47 gene sequences were obtained experimentally at NIH (GenBank accession numbers PV606815–PV606913) and University of North Carolina (GenBank accession numbers PV959330–PV959441) or retrieved from the publicly available databases of the Malaria Genomic Epidemiology Network, MalariaGEN (https://www.malariagen.net/data_package/open-dataset-plasmodium-vivax-v4-0/), PlasmoDB (https://plasmodb.org/plasmo/app/). the GenBank (https://www.ncbi.nlm.nih.gov/genbank/) and the European Nucleotide Archive (https://www.ebi.ac.uk/ena/browser/home). This publication uses data from the MalariaGEN Plasmodium vivax Genome Variation Project[34]. The 1199 Pv47 sequences used can be found in Supplementary Data 1, sequence ID indicates country of origin and source. Source data are provided with this paper.

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

## Acknowledgements

This work was supported by the Intramural Research Program of the Division of. Intramural Research, NIAID/NIH Z01AI000947 (C.B.-M.), and grant NIH/NIAID R01AI162947 (E.L.). The contributions of the NIH authors were made as part of their official duties as NIH federal employees, are in compliance with agency policy requirements, and are considered Works of the United States Government. However, the findings and conclusions presented in this paper are those of the authors and do not necessarily reflect the views of the NIH or the U.S. Department of Health and Human Services.

## Author contributions

A.M.-C., C.B.-M., L.G.-C., X.S., M.U.F., M.A.-H., S.H., E.L., J.C.S. designed research; A.M.-C., T.Z.B.T., N.R., M.Y., N.K., C.M., L.G.-C., A.D. and A.F. performed research; A.M.-C., A.D., C.B.-M., and J.C.S. analyzed data; A.M.-C. and C.B.-M. wrote the paper.

## Funding

## Competing interests

The authors declare no competing interests.

## Ethics statement

For the samples from Brazil, approval of the study protocol was obtained from the Ethical Review Board of the Institute of Biomedical Sciences of the University of São Paulo, Brazil (538/2004 and 773/2007). For the samples from Mexico, the institutional ethics committee of the National Institute of Public Health in Mexico approved protocols to obtain *P. vivax* infected blood samples from patients; protocol numbers CI-87 (CONACYT CB29005M), CI-297 (CONACYT CB31041-M) and CI-442 (CONACYT 2004-119–SALUD). Written informed consent was obtained from the parents or guardians of participating children or from adult participants.
