## [Peer Review file · Nature Communications]

Worldwide genetic diversity of *Plasmodium vivax* Pv47 is consistent with natural selection by anopheline mosquitoes

Corresponding Author: Dr Carolina Barillas-Mury

Version 0:

Reviewer comments:

Reviewer #1

(Remarks to the Author)

This manuscript explores the hypothesis that genetic diversity in the Pvs47 gene in *Plasmodium vivax* malaria parasites is mediated by selection for compatibility with transmission by different anopheline mosquito species across the globe. The authors present a population genetic analysis of Pvs47 from approximately 1200 publicly available *P. vivax* genome sequences, confirming and in some cases extending previous observations of population structure and evidence of selection. The authors also genotype Pvs47 in parasite-infected mosquitoes that were generated in controlled feeding experiments from malaria-infected individuals fed upon by *Anopheles albimanus* and *Anopheles pseudopunctipennis*. They find an amino acid polymorphism in Pvs47 that is associated with vector species.

This manuscript addresses an interesting topic using multiple data types, and the analyses are competently performed and clearly reported. The figures and tables are generally clear and interpretable, with some specific comments/questions noted below.

The impact of this manuscript could be strengthened if it went farther to explore some of the interesting hypotheses raised but not previously examined in the field. For example, in the Discussion, the authors raise the hypothesis that local diversity of vector species in Oceania could be driving the high Pvs47 diversity observed in parasites from that region, but do not perform a quantitative analysis of local malaria vector diversity vs. Pvs47 diversity. Similarly, the authors could test the hypothesis that clines in Pvs47 allele frequencies could be driven by the range boundaries of different anopheline vectors.

Why not also genotype Pvs47 in human feeding experiments that did not generate a strong enrichment effect in one vector species vs. the other? Perhaps there are haplotypes of Pvs47 in Mexico that facilitate infection by both *An. albimanus* and *An. pseudopunctipennis*? Or, if Pvs47 is undifferentiated in those two mosquito species in feeds from some individuals, perhaps that is evidence of other parasite loci that influence the penetrance of Pvs47 as a determinant of vector transmissibility. The pvs47 allelic association with vector species that the authors find is very strong, but given cited population structure between the highland and lowland Mexican parasite populations, presumably there are many other variants across the parasite genome that would also be differentiated between the vector species.

Specific comments:

The assertion in the introduction that *Pfalciparum* originated in central Africa requires a citation.

Line 152: The text here mentions 4,971 sequences examined, while the abstract says only 1,199?

Line 312: The classification of alleles as reference vs. non-reference would be more interesting if alleles were rather polarized as ancestral vs. derived, using the *P. vivax*-like genome sequences (and/or other more distantly related species in the *vivax* clade).

Line 335: Genbank rather than Genebank

Line 340: missing Genbank accession numbers

Figure 1 Panel B: the differing vertical axis ranges for Pf and Pv impair the ability to compare diversity between the parasite

species. Is this nonsynonymous diversity, or all nucleotide diversity? Is there a rationale for representing nucleotide diversity rather than amino acid diversity here? Given that some codons have multiple polymorphic positions, translating this diversity to the AA level could be more germane. Given that some nucleotide positions exhibit pi values greater than 0.5, this presumably implies that some of them are polyallelic rather than biallelic?

Reviewer #2

(Remarks to the Author)

Review of Molina-Cruz et al., Worldwide genetic diversity and population structure of *Plasmodium vivax* Pv47 1 is consistent with natural selection by anopheline mosquitoes

General comments The research reported in this manuscript extends a long term research program of this research team which resulted in a lock-and-key hypothesis for malaria ookinete parasite interactions with mosquito midgut receptors, mediated by so-called Pfs47 (of *Plasmodium falciparum*) and Pv47 (of *Plasmodium vivax*) surface proteins of the ookinete and particular receptors in the microvillar region of the adult *Anopheles* midgut epithelium. Additional prior research revealed that successful lock and key (receptor binding of the parasite ookinete and mosquito cell proteins) prevented an immune cascade involving a nitrification process that would obviate the establishment of an oocyst. This has led this research team to postulate that the Pfs47 and Pv47 proteins will be under strong selection in nature. It further has led them to postulate that there is natural variation in receptor binding amongst populations of the many different *Anopheles* species that carry these parasites. Here, they propose that this variation explains geographic patterns of *P. vivax* parasite association with particular species of *Anopheles* vectors, building upon recent experimental work. Although there is no new experimental work here, the investigators draw from recent published work with *P. vivax* strains and the two species *Anopheles pseudopunctipennis* and *Anopheles albimanus* to demonstrate associations between haplotype variants of *P. vivax* and infectivity in these species (or lack thereof). This study builds primarily an evolutionary analysis from sequence data, gene markers, molecular modeling, and inferences about selection. Importantly, the investigators demonstrate non-synonymous nucleotide substitutions at putative key effort domains of the PV47 protein, especially the D2 domain; and they note that the ratio of non-synonymous to synonymous substitutions is high, commonly a signal of positive selection. Overall, the authors build a strong case for their hypothesis of spread of *P. vivax* globally on the basis of compatibility of endemic mosquitoes with *P. vivax* haplotype variants. Their explanation will demand much more experimental work, which I think we can expect to be forthcoming.

Specific comments

1. Given the strong evidence for selection on Pv47 (and of course Pfs47, demonstrated here and addressed elsewhere by this research team), we should expect some kind of evolutionary arms race with the mosquito receptor protein but this publication contains no data on natural variation in that epithelial receptor. Is it evolving at a similar rate? What is the range of variation of its haplotypes in nature? Are there any evidences of mosquitoes that evolve incompatibility in receptor protein binding? How does this variation influence the immune response (for example, if the key doesn't fit the lock, is the nitration response immune always activated, or does the parasite simply not bind to the midgut epithelium cell and so the oocyst fails to establish for that reason?). This comment is not fatal to the study, but some discussion about whether the mosquito epithelial receptor protein is also under positive selection (and if it has other, perhaps constraining, natural functions) would be helpful here. A related question is the matter of cost to the vector and the parasite with regard to these signals of positive selection. If parasites evolve to retain infectivity and therefore prevent the mosquito immune response, then the cost to the parasite is clear; but the cost to the mosquito (loss of fitness) of infection of the malaria parasite is not so clear.
2. The "story" told here is really quite similar to Molina-Cruz et al. (2023) Role of Pfs47 in the dispersal of ancestral *Plasmodium falciparum* malaria through adaptation to different anopheline vectors. PNAS but that is fine. We should not automatically expect *P. vivax* populations to behave in the same way.
3. The authors may find the approach by Sjostrand et al. (2014) Private haplotypes can reveal local adaptation. BMC Genetics doi: 10.1186/1471-2156-15-61 useful because they explore the appearance of local or "private" haplotypes in populations as a signal of positive selection and provide a tool for analysis of their frequencies.
4. Line 250 no comma after dirus
5. Line 271 "evolutionary distant" should be "evolutionarily distant"
6. Line 193 should be "range" not "ranget"

Reviewer #3

(Remarks to the Author)

The *Plasmodium vivax* Pv47 gene is an ortholog of Pfs47 in *P. falciparum*, a gene that codes for a protein that binds *P. falciparum*'s oocysts to the midgut of *Anopheles* mosquitoes and also enables parasite evasion of the insect's immune response. As such, variations in Pfs47 genetic diversity, population structure among regions and vector compatibility are driven by different *Anopheles* species occupying different regions. In this paper, Molina-Cruz and colleagues extended their study on Pfs47 by sequencing and analyzing the genetic diversity and population structure of Pv47 isolates from different parts of the world to infer natural selection of the gene by malaria vectors. They also analyzed and compared amino acid mutations in Pv47 protein ligand in *P. vivax*-infected human blood samples that caused an infection in one vector species but not another in experimental infection studies.

Three key findings were reported in this paper. Firstly, there was an excess of non-synonymous mutations in a particular region of Pv47 that determines compatibility with mosquito species in its Pfs47 ortholog, indicating positive selection of Pv47 by vectors. Genetic diversity was high in regions with diverse vector species, indicating the role of mosquito species diversity in driving the evolution of the gene. Secondly, the gene exhibited a strong regional population structure consistent with variation in vector species among the regions, further supporting the hypothesis of vector-driven natural selection of the gene. Third, they show that in Chaiapas, Mexico, *P. vivax* from human blood samples that infected *Anopheles albimanus* more efficiently than *An. pseudopunctipennis* had a non-synonymous amino acid mutation that was different from those samples that efficiently infected *An. pseudopunctipennis* more than *An. albimanus*, further supporting the hypothesis of vector-driven evolution of vector compatibility of Pv47.

The paper was well written with sufficient presentation of context. The methodology, including data analysis were sound and appropriate and the results are strong. The conclusions presented were consistent with the results. The detail presented are sufficient, which allow this study to be reproduced by other investigators. In my view, the work is of good quality, generally met the standard in the field of molecular parasitology and is worthy of publication in Nature Communications.

Version 1:

Reviewer comments:

Reviewer #1

(Remarks to the Author)

I am satisfied with the authors' responses to my previous comments, although I believe the impact of this manuscript would be greater if its scope were broadened.

Response to reviewers

The authors appreciate the reviewers' comments and insights, they are contributing to improve the manuscript. Please find below a detailed response to the points raised by the reviewers.

Reviewer #1 (Remarks to the Author):

This manuscript explores the hypothesis that genetic diversity in the Pvs47 gene in Plasmodium vivax malaria parasites is mediated by selection for compatibility with transmission by different anopheline mosquito species across the globe. The authors present a population genetic analysis of Pvs47 from approximately 1200 publicly available P. vivax genome sequences, confirming and in some cases extending previous observations of population structure and evidence of selection. The authors also genotype Pvs47 in parasite-infected mosquitoes that were generated in controlled feeding experiments from malaria-infected individuals fed upon by Anopheles albimanus and Anopheles pseudopunctipennis. They find an amino acid polymorphism in Pvs47 that is associated with vector species.

This manuscript addresses an interesting topic using multiple data types, and the analyses are competently performed and clearly reported. The figures and tables are generally clear and interpretable, with some specific comments/questions noted below.

The impact of this manuscript could be strengthened if it went farther to explore some of the interesting hypotheses raised but not previously examined in the field. For example, in the Discussion, the authors raise the hypothesis that local diversity of vector species in Oceania could be driving the high Pvs47 diversity observed in parasites from that region, but do not perform a quantitative analysis of local malaria vector diversity vs. Pvs47 diversity. Similarly, the authors could test the hypothesis that clines in Pvs47 allele frequencies could be driven by the range boundaries of different anopheline vectors.

RESPONSE: The authors appreciate the interesting suggestions. We have an ongoing collaboration to study the genetic diversity of Pfs47, and eventually Pv47, in Papua New Guinea parasite populations, to try establish if different haplotypes correlate it with local vector species. One would have to consider both vector diversity and phylogeny, and ideally sequence the P47 receptor of the different vectors. This is work in progress beyond the scope of the current manuscript, and we currently do not have enough data to make any conclusions. We are interested in correlating gradual changes in Pv47 haplotype frequency with vector boundaries. That is the reason why we looked at the Mexican Pv47 population in this manuscript. In that case, there is a correlation between a Pv47 polymorphism and the transmission by two different vectors whose ranges are contiguous in the state of Chiapas (*An. albimanus* vs *An. pseudopunctipennis*).

Why not also genotype Pvs47 in human feeding experiments that did not generate a strong enrichment effect in one vector species vs. the other? Perhaps there are haplotypes of

Pvs47 in Mexico that facilitate infection by both *An. albimanus* and *An. pseudopunctipennis*? Or, if *Pvs47* is undifferentiated in those two mosquito species in feeds from some individuals, perhaps that is evidence of other parasite loci that influence the penetrance of *Pvs47* as a determinant of vector transmissibility. The *pvs47* allelic association with vector species that the authors find is very strong, but given cited population structure between the highland and lowland Mexican parasite populations, presumably there are many other variants across the parasite genome that would also be differentiated between the vector species.

RESPONSE: The reviewer raises a valid point regarding whether there were *P. vivax* isolates that infected both vectors to similar levels. We first analyzed the results from all previous infections conducted in Mexico and eliminated those infections with less than 5 blood-fed mosquitoes in at least one of the vectors, and those with an infection prevalence of 0.2 or less for both vectors (shown in gray in the table below). Genomic DNA samples from infected patients were used to amplify and sequence Pv47 and we reported the results from all the samples that could be sequenced (some genomic DNA samples were degraded and the PCR reactions failed and could not be analyzed). We also determined if there was a significant difference in infection prevalence between the two vectors in all the samples that were sequenced. We found that all samples exhibited a significant difference in the prevalence of infection, with significance values ranging from $*(P<0.05)$, $** (P<0.01)$ to $*** (P<0.001)$, mostly influenced by the number of blood-fed mosquitoes in the samples (infections with more blood-fed females having stronger statistical significances) (see table below). Even in the samples with less than 5 infected mosquitoes that were excluded, there were marked differences in the average number of oocysts present in the two mosquito species.

We can't eliminate the possibility that there are other *P. vivax* genes determining vector compatibility besides Pv47. The Pv47 polymorphism that correlates with differential vector infectivity (K27E) does not align exactly with the *P. vivax* population structure identified using microsatellite markers by Joy et al. While all C1 parasites from the coastal area carry the K27 allele and resulted in higher infection in *An. albimanus*, both F1 and F2 *P. vivax* populations (from higher altitudes) share the same Pv47 alternate (E27) allele. This suggests that if other parasite loci determine vector compatibility, those loci must be shared between F1 and F2. However, this is beyond the scope of our study. We have now added the subpopulation classification of each isolate to Table S6, and a comment clarifying this point has been added in the results section of the revised manuscript.

Specific comments:

The assertion in the introduction that *P. falciparum* originated in central africa requires a citation.

RESPONSE: Thank you for the suggestion, the citation has been added.

Line 152: The text here mentions 4,971 sequences examined, while the abstract says only 1,199?

RESPONSE: There were 4,971 *P. falciparum* Pfs47 sequences and 1,199 *P. vivax* Pv47 sequences analyzed. This has been clarified in the revised manuscript.

Line 312: The classification of alleles as reference vs. non-reference would be more interesting if alleles were rather polarized as ancestral vs. derived, using the *P. vivax*-like genome sequences (and/or other more distantly related species in the vivax clade).

RESPONSE: We agree with the reviewer that identifying ancestral alleles would be more informative. The following paragraph has been added to the discussion, together with new Fig. S4 and S5 in the Supplemental section.

“The *P. vivax*-like parasites that infect primates in central Africa are the closest known relatives to *P. vivax*. An analysis of 10 *P. vivax*-like P47 sequences recovered from chimpanzees and one mosquito³² revealed higher sequence similarities to Pv47 haplotypes circulating in Africa and Oceania (haplotype “e” in Fig S4; and red circles in Fig. S5), including in polymorphisms with strong population structure in Pv47 ($F_{st} > 0.2$; Fig. S4). This suggests that the ancestral *P. vivax*, presumably transmitted by sylvatic ape malaria vectors (e.g., *An. moucheti*), could have readily adapted to human malaria vectors in Africa and Oceania.”

Line 335: Genbank rather than Genebank

RESPONSE: Thank you for the observation, the correction has been done.

Line 340: missing Genbank accession numbers

RESPONSE: The Pv47 sequences have been submitted to the Genbank. The accession numbers are now included in the text of the revised manuscript.

Figure 1 Panel B: the differing vertical axis ranges for Pf and Pv impair the ability to compare

diversity between the parasite species. Is this nonsynonymous diversity, or all nucleotide diversity? Is there a rationale for representing nucleotide diversity rather than amino acid diversity here? Given that some codons have multiple polymorphic positions, translating this diversity to the AA level could be more germane. Given that some nucleotide positions exhibit pi values greater than 0.5, this presumably implies that some of them are polyallelic rather than biallelic?

RESPONSE: The vertical axis for nucleotide diversity (P_i) values in Fig. 1B is now in the same scale for both Pv47 and Pfs47. The diversity shown is total nucleotide diversity (synonymous and non-synonymous substitutions) but most polymorphisms (47 out of 71) are non-synonymous. All sites with $P_i > 0.05$ are non-synonymous, i.e. the graphs are very similar if we only include non-synonymous polymorphisms. We have included a sentence (page 6 line118) that reads: “Sites

with nucleotide diversity (π) >0.05 were found to be non-synonymous substitutions (Fig. 1B)". .
There is only one site (T785A/C) that is polyallelic (Table S2).

Reviewer #2 (Remarks to the Author):

Review of Molina-Cruz et al., Worldwide genetic diversity and population structure of Plasmodium vivax Pv47 1 is consistent with natural selection by anopheline mosquitoes

General comments:

The research reported in this manuscript extends a long-term research program of this research team which resulted in a lock-and-key hypothesis for malaria ookinete parasite interactions with mosquito midgut receptors, mediated by so-called Pfs47 (of Plasmodium falciparum) and Pv47 (of Plasmodium vivax) surface proteins of the ookinete and particular receptors in the microvillar region of the adult Anopheles midgut epithelium. Additional prior research revealed that successful lock and key (receptor binding of the parasite ookinete and mosquito cell proteins) prevented an immune cascade involving a nitrification process that would obviate the establishment of an oocyst. This has led this research team to postulate that the Pfs47 and Pv47 proteins will be under strong selection in nature. It further has led them to postulate that there is natural variation in receptor binding amongst populations of the many different Anopheles species

that carry these parasites. Here, they propose that this variation explains geographic patterns of P. vivax parasite association with particular species of Anopheles vectors, building upon recent experimental work. Although there is no new experimental work here, the investigators draw from recent published work with P. vivax strains and the two species Anopheles pseudopunctipennis and Anopheles albimanus to demonstrate associations between haplotype variants of P. vivax and infectivity in these species (or lack thereof). This

study builds primarily an evolutionary analysis from sequence data, gene markers, molecular modeling, and inferences about selection. Importantly, the investigators demonstrate non-synonymous nucleotide substitutions at putative key effort domains of the PV47 protein, especially the D2 domain; and they note that the ratio of non-synonymous to synonymous substitutions is high, commonly a signal of positive selection.

Overall, the authors build a strong case for their hypothesis of spread of P. vivax globally on

the basis of compatibility of endemic mosquitoes with P. vivax haplotype variants. Their explanation will demand much more experimental work, which I think we can expect to be forthcoming.

Specific comments

1. Given the strong evidence for selection on Pv47 (and of course Pfs47, demonstrated here and addressed elsewhere by this research team), we should expect some kind of evolutionary arms race with the mosquito receptor protein but this publication contains no

data on natural variation in that epithelial receptor. Is it evolving at a similar rate? What is the range of variation of its haplotypes in nature? Are there any evidences of mosquitoes that evolve incompatibility in receptor protein binding? How does this variation influence the immune response (for example, if the key doesn't fit the lock, is the nitration response immune always activated, or does the parasite simply not bind to the midgut epithelium cell and so the oocyst fails to establish for that reason?). This comment is not fatal to the study, but some discussion about whether the mosquito epithelial receptor protein is also under positive selection (and if it has other, perhaps constraining, natural functions) would be helpful here. A related question is the matter of cost to the vector and the parasite with regard to these signals of positive selection. If parasites evolve to retain infectivity and therefore, prevent the mosquito immune response, then the cost to the parasite is clear; but the cost to the mosquito (loss of fitness) of infection of the malaria parasite is not so clear.

RESPONSE: The reviewer brings very interesting questions, that we have tried to address previously. To our knowledge, there is no evidence of amino acid (AA) polymorphisms in the Pfs47 receptor (P47Rec) in a given *Anopheles* species (e.g. *An. gambiae*). In fact, the AA sequence of the P47Rec is identical in most members of the *Anopheles gambiae* complex (Molina-Cruz, A. *et al.* 2020, *PNAS*, 201917042, doi:10.1073/pnas.1917042117). That is, there is no evidence of positive selection of the P47Rec. This suggests that the mosquito is not evolving incompatibility to the parasite at the level of the Pfs47 Receptor.

Even in highly endemic areas only 1-5% of mosquitoes are infected with *P. falciparum*, so it is likely that the parasite doesn't exert enough selective pressure in the mosquito population. The fitness cost of *Plasmodium* infection to the vector is not clear. Some infections using model systems with non-natural parasite-vector combinations (e.g. *P. yoeli* infection of *An. gambiae*) show a fitness cost to the vector but no selection of resistance to infection in the mosquito (Hurd, H. *et al.*, 2005 *Evolution* **59**, 2560-2572). However, some carefully designed studies with compatible vector-parasite combinations (*P. falciparum* infection of *An. gambiae*) found that *Plasmodium* infection reduced survival only in mosquito strains selected for insecticide resistant but not in the susceptible strain. Furthermore, infection was associated with an increase in fecundity independently of the strain considered (Alout *et al.*, 2016 *Sci Rep* 6, 29755).

2. The "story" told here is really quite similar to Molina-Cruz *et al.* (2023) Role of Pfs47 in the dispersal of ancestral *Plasmodium falciparum* malaria through adaptation to different anopheline vectors. *PNAS* but that is fine. We should not automatically expect *P. vivax* populations to behave in the same way.

RESPONSE: We agree with the reviewer. *P. vivax* is evolutionarily distant to *P. falciparum*, at least as distant as mouse malaria parasites like *P. berghei* is from *P. falciparum*. This is illustrated by the fact that Pv47 has only 43% identity to Pfs47 at the AA level. Furthermore, *P. vivax* and *P. falciparum* also exhibit important differences in their biology, for example, *P. vivax* can lead to persistent liver infection by forming hypnozoites and they can only infect reticulocytes, so it was not clear *a priori* if they shared a common strategy to evade the mosquito immune system. However, the fact that Pv47 shares all the genetic diversity features studied (e.g. location of polymorphisms, population structure) with Pfs47 is consistent with a common selective pressure driven by natural selection of parasite populations, when they encounter evolutionarily distant mosquitoes as infected humans migrate to new geographic regions.

3. The authors may find the approach by Sjostrand et al. (2014) Private haplotypes can reveal local adaptation. BMC Genetics doi: 10.1186/1471-2156-15-61 useful because they explore the appearance of local or "private" haplotypes in populations as a signal of positive selection and provide a tool for analysis of their frequencies.

RESPONSE: We appreciate the comment. The Maximum Frequency of Private Haplotypes (MFPH) approach mentioned by Sjostrand et al. seems to perform similarly to Fst. It seems to the authors that in the case of a single gene sequence like Pv47, under strong natural selection, the analysis of Fst at nucleotide level provides enough resolution to identify SNPs under selection. .

4. Line 250 no comma after dirus

RESPONSE: Thank you, the typo has been corrected.

5. Line 271 "evolutionary distant" should be "evolutionarily distant"

RESPONSE: Thank you, it has been corrected.

6. Line 193 should be "range" not "ranget"

RESPONSE: Thank you, it has been corrected.

Reviewer #3 (Remarks to the Author):

The Plasmodium vivax Pv47 gene is an ortholog of Pfs47 in P. falciparum, a gene that codes for a protein that binds P. falciparum's oocysts to the midgut of Anopheles mosquitoes and also enables parasite evasion of the insect's immune response. As such, variations in Pfs47 genetic diversity, population structure among regions and vector compatibility are driven by different Anopheles species occupying different regions. In this paper, Molina-Cruz and colleagues extended their study on Pfs47 by sequencing and analyzing the genetic diversity and population structure of Pv47 isolates from different parts of the world to infer natural selection of the gene by malaria vectors. They also analyzed and compared amino acid mutations in Pv47 protein ligand in P. vivax-infected human blood samples that caused an infection in one vector species but not another in experimental infection studies.

Three key findings were reported in this paper. Firstly, there was an excess of non-synonymous mutations in a particular region of Pv47 that determines compatibility with mosquito species in its Pfs47 ortholog, indicating positive selection of Pv47 by vectors. Genetic diversity was high in regions with diverse vector species, indicating the role of mosquito species diversity in driving the evolution of the gene. Secondly, the gene exhibited a strong regional population structure consistent with variation in vector species among the regions, further supporting the hypothesis of vector-driven natural selection of the gene. Third, they show that in Chiapas, Mexico, P. vivax from human blood samples that infected Anopheles albimanus more efficiently than An. pseudopunctipennis had a non-synonymous amino acid mutation that was different from those samples that efficiently infected An. pseudopunctipennis more than An. albimanus, further supporting the hypothesis of vector-driven evolution of vector compatibility of Pv47.

The paper was well written with sufficient presentation of context. The methodology, including data analysis were sound and appropriate and the results are strong. The conclusions presented were consistent with the results. The detail presented are sufficient, which allow this study to be reproduced by other investigators. In my view, the work is of good quality, generally met the standard in the field of molecular parasitology and is worthy of publication in Nature Communications.

RESPONSE: The authors appreciate the comments.